# Correlations of Cross-Entropy Loss in Machine Learning

**DOI:** 10.3390/e26060491

**Published:** 2024-06-03

**Authors:** Richard Connor, Alan Dearle, Ben Claydon, Lucia Vadicamo

**Affiliations:** 1School of Computer Science, University of St Andrews, St Andrews KY16 9SS, UK; al@st-andrews.ac.uk (A.D.); bc89@st-andrews.ac.uk (B.C.); 2Institute of Information Science and Technologies, Italian National Research Council (CNR), 56124 Pisa, Italy; lucia.vadicamo@isti.cnr.it

**Keywords:** softmax, cross-entropy, f-divergence, Kullback–Leibler divergence, Jensen–Shannon divergence, triangular divergence

## Abstract

Cross-entropy loss is crucial in training many deep neural networks. In this context, we show a number of novel and strong correlations among various related divergence functions. In particular, we demonstrate that, in some circumstances, (a) cross-entropy is almost perfectly correlated with the little-known triangular divergence, and (b) cross-entropy is strongly correlated with the Euclidean distance over the logits from which the softmax is derived. The consequences of these observations are as follows. First, triangular divergence may be used as a cheaper alternative to cross-entropy. Second, logits can be used as features in a Euclidean space which is strongly synergistic with the classification process. This justifies the use of Euclidean distance over logits as a measure of similarity, in cases where the network is trained using softmax and cross-entropy. We establish these correlations via empirical observation, supported by a mathematical explanation encompassing a number of strongly related divergence functions.

## 1. Introduction

The notion of cross-entropy loss is integral to the training of many deep learning networks. Before the cross-entropy loss function can be applied to arrays within the network, a softmax function is normally applied in order to convert an array of arbitrary floating point values into an array of strictly positive values which sum to 1.

The softmax function has a single hyper-parameter, temperature, which governs the input to the cross-entropy function. Until recently, this value had not been rigorously investigated. Recent work [1] has shown that a wide range of values can be useful, typically in the range 0.1 to 100.

In this study, we demonstrate strong correlations, as shown in Figure 1, between cross-entropy, triangular divergence, and Euclidean divergence (see Table 1) in the context of machine learning. These correlations are particularly strong when higher temperature values are used within the softmax function.

The main contribution of this article is to show:A very tight correlation among all the information divergence functions, i.e., the cross-entropy divergence (CED), Kullback–Leibler divergence (KLD), Jensen–Shannon divergence (JSD), and triangular divergence (TRI) for spaces with certain properties, along with the demonstration that the output of many deep learning networks have these properties;A tight correlation between the Euclidean divergence (EUC) over the logit space and CED to which the softmax function has been applied with a high temperature.

The effects we measure are found in high-dimensional data; they are probabilistic and therefore beyond detailed mathematical analysis. To this extent, our results are empirical; however, we show that they are highly repeatable, and we provide a significant mathematical explanation of why they occur.

For the sake of accuracy and absolute clarity, we proceed to give formal definitions of the functions we refer to in the text:$$\begin{array}{cccc} h(x)\; & =\;-x\log_{2}x & \; & \end{array}$$(1)$$\begin{array}{cccc} softmax(x,t)\; & =\;\frac{1}{\sum_{i=1}^{n}e^{x_{i}/t}}\;\left[e^{x_{1}/t},\ldots ,e^{x_{n}/t}\right] & \; & \mathrm{Softmax} \end{array}$$(2)$$\begin{array}{cccc} CED(q:p)\; & =\;-\sum_{i=1}^{n}q_{i}\log p_{i} & \; & \mathrm{Cross}-\mathrm{entropy}\; \end{array}$$(3)$$\begin{array}{cccc} KLD(q:p)\; & =\;\sum_{i=1}^{n}q_{i}\log\frac{q_{i}}{p_{i}} & \; & \mathrm{Kullback}–\mathrm{Leibler}\;\mathrm{div}. \end{array}$$(4)$$\begin{array}{cccc} JSD(q,p)\; & =\;1-\frac{1}{2}\sum_{i=1}^{n}h\left(q_{i}\right)+h\left(p_{i}\right)-h\left(q_{i}+p_{i}\right) & \; & \mathrm{Jensen}–\mathrm{Shannon}\;\mathrm{div}. \end{array}$$(5)$$\begin{array}{cccc} TRI(q,p)\; & =\;\frac{1}{2}\sum_{i=1}^{n}\frac{{\left(q_{i}-p_{i}\right)}^{2}}{q_{i}+p_{i}} & \; & \mathrm{Triangular}\;\mathrm{divergence} \end{array}$$(6)$$\begin{array}{cccc} EUC(q,p)\; & =\;\sum_{i=1}^{n}{\left(q_{i}-p_{i}\right)}^{2} & \; & \mathrm{Euclidean}\;\mathrm{divergence} \end{array}$$

We use the notation $q_{i},p_{i}$ to refer to the individual (component) dimensions of *n*-dimensional vectors q,p. We use the notation F(q:p) to define a divergence function which may not be symmetric over its arguments and F(q,p) to denote a symmetric function. The JSD and TRI are normalised so that their outcome is in [0,1]. Other than the CED, an outcome of 0 implies q=p. Post-softmax, all values $q_{i},p_{i}$ are in the range (0,1), so all functions are always well defined.

The correlations we establish here are interesting in their own right, and also have two possible practical applications. First, we note that if the CED is perfectly correlated with the TRI, there exists a simple re-written form of the TRI (see Equation (Equation 23)), which as we show is a much cheaper function to evaluate, potentially allowing the saving of many compute cycles during training.

Second, for some types of network, if the EUC over the logit space is perfectly correlated with the CED over the softmax equivalent, this seems to imply that Euclidean distance over the same space post-training should be the metric of choice for assessing similarity. Common practice seems to usually recommend cosine distance for this purpose.

The rest of this article is structured as follows. Section 2.1 gives an overview of the use of loss functions in the training of neural networks and introduces the concepts of the softmax function and its temperature parameter. Section 3 introduces the experimental datasets we use and some other important aspects of our methodology. Section 4, Section 5, Section 6, Section 7, Section 8 and Section 9 show details of the individual correlations noted. Finally, we discuss some of the outcomes in Section 10 before concluding in Section 11.

## 2. Background and Related Work

### 2.1. Neural Networks and Loss Functions

We are interested in the application of information loss functions in the context of the training of deep neural networks. For our purposes, we largely treat a network as a “black box” as described below.

A neural network may be abstractly represented by a function *f*, which takes as input some value representation x and a set of parameters (weights) θ and return an output f(x,θ). Deep neural networks are typically organised as a sequence (or graph) of parametric transformations whose composition gives the final function f(·). The parameters of the network are learned (optimised) during the training phase to minimise a loss (or cost) function measuring the discrepancy between the network’s outputs and target values. In particular, given a training set of *N* input–target pairs $X={\left\{\left(x^{\left(i\right)},y^{\left(i\right)}\right)\right\}}_{i=1}^{N}$, the quality of a particular configuration of parameters is quantitatively assessed by a loss function L(X,θ). It is important to note that the training data comprise samples $x^{\left(i\right)}$ drawn from the real data distribution, with target output values $y^{\left(i\right)}$ typically obtained through manual annotation or directly derived from the inputs $x_{i}$, as seen in self-supervised methods [2]. The training process involves iteratively adjusting the parameters θ to minimise the expected loss over the training data, relying on the assumption that the training set is large enough to represent some truth encompassing all future inputs to the network. The particular formulation of the loss function is task-dependent. The essential requirements for the loss function L are (1) that a smaller loss represents a stronger similarity between the output of the network and the target output and (2) that it is differentiable, in order to feed back to the process of making appropriate adjustments to θ between iterations. Here, our focus lies on scenarios where a cross-entropy loss (applied to the outputs produced by a softmax function) is employed, a common approach found in several state-of-the-art neural networks.

Formally, we are considering the case in which f(x,θ)=softmax(z(x,θ),t) where z(x,θ) are the logits (pre-softmax output of the network) and *t* is the temperature used in the softmax. Please note that z(x,θ) takes as input some value representation x and a set of weights θ and returns a vector of floating point values. Cross-entropy divergence (Equation (2)) is defined over a finite set of probabilities; therefore, before it can be applied, the logit vectors must be converted to a vector of positive numbers which sum to 1. To preserve the *argmax* property, the conversion must also maintain the position of the largest value within the vector. The conversion is typically performed using the softmax function (Equation (Equation 1)).

The cross-entropy loss can be written as
(7)$$\begin{array}{cc} L(X,\theta ) & =\sum_{(x,y)\in X}CED\left(y:f\left(x,\theta \right)\right)\\ \; & =-\sum_{(x,y)\in X}y\cdot \log\left(\mathrm{softmax}(z(x,\theta ),t)\right)\\ \; & =-\sum_{(x,y)\in X}\sum_{i=1}^{n}y_{i}\log\left(\frac{e^{z_{i}/t}}{\sum_{j=1}^{n}e^{z_{j}/t}}\right) \end{array}$$
where, for simplicity, $z_{j}$ denotes the *j*-th element of the logit vector z(x,θ).

Note that the term $e^{z_{i}/t}$ in the softmax is monotonically increasing with $z_{i}$, and the denominator ($\sum_{i}e^{z_{i}/t}$) simply performs an $L_{1}$ normalization of the outcome. Agarwala et al. [1] state that softmax followed by cross-entropy is “a principled approach to modelling probability distributions”. Softmax is essentially a differentiable *argmax* function, as required for training purposes [3]. However, it shows an arbitrary non-linearity depending on the range of values applied as exponents and the value of the temperature parameter *t*.

### 2.2. Softmax Temperature

The use of different temperatures within the softmax function has two major effects. First, a higher temperature gives more significance to smaller values within the logit vectors; low temperatures have the effect of allowing the larger values to dominate the ensuing comparisons. Secondly, higher temperatures also result in inputs to the cross-entropy function which have a decreased *measure of roughness* (see Section 6.1); in short, the variance among the dimensions is decreased, thus increasing the entropy. The effect of this has been previously studied in the context of various information divergence functions [4,5], and underlies the strong correlations we report in this article.

Figure 2 shows the effect of temperature on $e^{x}$ on the softmax function application. It can be seen that when a relatively low temperature is used (e.g., t=1), for low negative values, the function maps to almost zero and quickly maps to very high values as *x* is increased through zero and into the positive domain. By contrast, with a high temperature (t=10), the domain of the function more gently rises through the shown range, and with t=100 the function becomes effectively linear.

A major observation shown in [1], that a higher temperature leads to semantically better results, while a lower temperature leads to faster convergence of the network, seems consistent with this observation. The main result expressed in [1] is that the best temperature is very dependent upon context, but nonetheless a wide range of temperatures may be useful, perhaps with different temperatures at different stages of training. They suggest using temperatures in the range 0.1 up to 100. We hypothesise that an appropriate of choice of temperature selected according to the value range in the logits can result in a more efficacious loss function.

### 2.3. f-Divergences

In mathematics, an *f*-divergence is any numeric function which allocates a value to the dissimilarity between two sets of probabilities. In our context, this includes the CED, KLD, JSD, and TRI. It should be noted however that, mathematically, these are all defined over sets of probabilities, whereas in our case we apply them to vectors of positive numbers which sum to 1. That is, the application of the softmax function to an arbitrary vector of floating point numbers is not actually a probability distribution. Rather, softmax has been defined as a somewhat arbitrary function which maps floating point vectors to a domain where an *f*-divergence can be used as a loss function.

We rely heavily on work by Harremoës [5] and Topsøe [4], who show strong bounds among these functions; we extend that work here to show how these bounds give extremely strong correlations among high-dimensional embeddings. Other more recent related work includes [6], which shows a very strong convergence of the measured distribution of values as the locality over which the distances are measured tends towards infinitesimal.

In the rest of this article, we show some very strong correlations between cross-entropy loss over high-temperature softmax conversions and a number of other loss functions. These are, we believe, interesting for their own sake. Furthermore, they may help to guide practitioners in the field of neural networks to a more informed choice of temperature according to properties of the logit vectors being produced by the network.

## 3. Methodology

For the experiments, we used logits deriving from the following deep learning networks: GoogleNet [7] trained on Places365 classifications [8]; SqueezeNet [9] and AlexNet [10] trained on ImageNet classifications [11]; and DinoV2 [12] outputs. In all cases, we derived logits from the first 10,000 images of the MirFlickr one million image set [13]. These data are summarised in Table 2. All of our (MATLAB) code and data are available at https://github.com/MetricSearch/2024_entropy_paper (accessed on 29 May 2024).

To calculate correlations, we used the Spearman rho correlation function [14], a topological measure of the order preservation of divergence within sampled pairs of objects from the domain. This is essentially a measure of the likelihood for functions *f* and *g* that f(x,y)<f(x,z) implies g(x,y)<g(x,z). We use this form of the function:
(8)$$S_{\rho }=1-\frac{6\sum_{i=1}^{T}{\left(z\left(i\right)-\hat{z}\left(i\right)\right)}^{2}}{T^{3}-T}$$
where z(i) and $\hat{z}\left(i\right)$ are the values obtained by *f* and *g* over a set of *T* function applications. The adjusting factors combine to give an output in the range [−1,1], where 1 implies a perfect preservation of ordering, 0 implies no correlation, and −1 implies a perfect inverse correlation.

We also give visual impressions of correlations using Shepard diagrams [15]. These are scatter plots of one divergence function against the other over a finite set of samples, decorated with the isotonic regression function defined for the Kruskal stress coefficient [16]. They give a useful visual impression of correlation. Shepard diagrams are normally annotated with the Kruskal stress value; however, this is dependent on the absolute range of only one of the functions. As the absolute ranges vary hugely among the different functions we tested, this would give incomparable results, and we therefore report Spearman rho values instead.

Figure 3 gives examples of two simple Shepard plots demonstrating these for perfectly correlated, and highly correlated, functions.

There are two important points to note about the graphs we present.

As the softmax temperatures are increased, and therefore all values within the vectors under consideration converge, the absolute distances yielded by all the information divergence functions become very similar; however, the rankings are still perfectly significant. This is why all of our analysis was performed using correlations, rather than any other measure.In all of our comparisons, we arbitrarily choose a single element and measure it against a large number of different elements. This is always appropriate for our context, but note that the correlations measured in this context are quite different from those that would be measured if both arguments were randomly selected. There is a danger with this methodology that the choice of single element may be atypical; in all cases, we have repeated these experiments with many different elements, not shown for brevity, to ensure the results presented are general.

## 4. Correlation of Cross-Entropy and Triangular Divergence

Figure 4 shows the correlation between cross-entropy and triangular divergence applied to the GoogleNet-places network [7,8]. Both functions are applied to the values after softmax using a range of temperatures. As can be seen, temperatures of around 10 and upwards lead to an almost perfect correlation.

The steps to demonstrating the underlying reasons for this correlation are as follows:The CED is a specialised form of the KLD, such that CED(k:p) perfectly correlates with KLD(k:p) for all probability vectors p compared with a fixed vector k.The relationship between the KLD and JSD appears evident, yet it is influenced by the temperature setting in the softmax function. Notably, while strong correlation exists at higher temperatures, lower temperatures exhibit a diminished correlation.Jensen–Shannon divergence correlates almost perfectly with triangle divergence in almost all high-dimensional spaces.From all the above, cross-entropy divergence correlates very strongly with triangular divergence with higher temperature values. We note that triangular divergence is a much cheaper calculation than cross-entropy, and if the correlation is very strong the latter may be used instead.

We show each of these steps in turn.

## 5. Correspondence between Cross-Entropy and Kullback–Leibler Divergence

The perfect correlation between cross-entropy and Kullback–Leibler divergence is well known and derives from simple algebra:(9)$$KLD\left(q:p\right)=\sum_{i=1}^{n}q_{i}\log\frac{q_{i}}{p_{i}}=\sum_{i=1}^{n}\left(q_{i}\log q_{i}-q_{i}\log p_{i}\right)=CED\left(q:p\right)-H\left(q\right)$$
where $H\left(q\right)=-\sum_{i=1}^{n}q_{i}\log q_{i}$ is the Shannon entropy of q. If q is fixed, its Shannon entropy remains constant, ensuring a perfect correlation. It is important to note that both functions are asymmetric, and this perfect correlation applies only when a set of different probabilities {p} are compared with a single, fixed-value q supplied as the first parameter; otherwise, the correlation does not hold. In the context of comparing the information loss between a target output and a neural network model distribution, as described in Section 2.1, this will always be the case during supervised training of a classification network but depends on the network architecture for other types. Specifically, using the notation in Section 2.1, during neural network training, the vector q corresponds to the target output y and p to softmax(z(x,θ),t) for any input–target pair (x,y) of the training set. In the context of supervised classification, the target outputs y for a given input x remain fixed during training steps; consequently, the parameters θ that minimise KLD(y:softmax(z(x,θ),t)) are identical to those that minimise the loss CED(y:softmax(z(x,θ),t) since the two functions differ by a constant.

## 6. Correlation between Kullback–Leibler Divergence and Jensen–Shannon Divergence

Jensen–Shannon divergence, also called capacitory discrimination in the literature [4,17], derives from Kullback–Leibler divergence and is widely regarded as a bounded, smoothed, and symmetrised version of that:(10)$$JSD\left(q,p\right)=KL\left(q:\frac{q+p}{2}\right)+KL\left(p:\frac{q+p}{2}\right)$$
where $\frac{q+p}{2}$ is the mixture distribution of q and p. Therefore, the JSD can be interpreted as the total divergence to the average distribution $\frac{q+p}{2}$ [18].

It is noted in [4] and elsewhere that this is equivalent to an expression over the entropy *H* of the terms q and p:(11)$$JSD\left(q,p\right)=2H\left(\frac{q+p}{2}\right)-H\left(p\right)-H\left(q\right)$$
which is
(12)$$JSD\left(q,p\right)=\sum_{i}q_{i}\log q_{i}+p_{i}\log p_{i}-\left(q_{i}+p_{i}\right)\log\left(\frac{q_{i}+p_{i}}{2}\right)$$
and which then simplifies (using base 2 logs to simplify the constant term) to
(13)$$JSD\left(q,p\right)=2+\sum_{i}q_{i}\log q_{i}+p_{i}\log p_{i}-\left(q_{i}+p_{i}\right)\log\left(q_{i}+p_{i}\right)$$
In this context we can take 0log0=0, as xlogx tends to 0 from above as *x* does. We note also that the summand is equal to $2p_{i}$ if and only if $q_{i}=p_{i}$. This form can then be seen to give an outcome in [0,2], with 2 for orthogonal inputs (i.e., for all terms $q_{i}=0\vee p_{i}=0$) and 0 if q=p, so to normalise the output into [0,1] we use
(14)$$\begin{array}{cc} JSD(q,p) & =1+\frac{1}{2}\sum_{i}q_{i}\log q_{i}+p_{i}\log p_{i}-\left(q_{i}+p_{i}\right)\log\left(q_{i}+p_{i}\right) \end{array}$$
(15)$$\begin{array}{cc} \; & =1-\frac{1}{2}\sum_{i}h\left(q_{i}\right)+h\left(p_{i}\right)-h\left(q_{i}+p_{i}\right) \end{array}$$
which is the form given in Equation ().

With high temperature values, we measure almost perfect correlations between the KLD and JSD. Figure 5 shows Shepard diagrams of the KLD to JSD over AlexNet, GoogleNet, SqueezeNet, and DinoV2 data with softmax temperatures of 1, 10, and 100, respectively.

It is clear from its derivation that there is a strong semantic relationship between the KLD and JSD, but this alone does not explain the very strong correlations shown in the figure. We shed some light on the mathematical underpinnings in the next section.

### 6.1. Index of Coincidence and the Measure of Roughness

In [5], the authors introduce the notions of Index of Coincidence (IC) and the consequent measure of roughness (MR). The concepts are simple, giving measures essentially for the uniformity of terms within a set of probabilities:(16)$$\begin{array}{c} IC\left(p\right)=\sum_{i=1}^{n}p_{i}^{2} \end{array}$$

The underlying intuition of the MR measure is that of a divergence from the “flattest” set of probabilities, i.e., $U_{n}=\left[1/n,\ldots ,1/n\right]$:(17)$$\begin{array}{c} MR\left(p\right)=\sum_{i=1}^{n}{\left(p_{i}-\frac{1}{n}\right)}^{2} \end{array}$$
An alternative formulation of MR is
(18)$$\begin{array}{c} MR\left(p\right)=\frac{1}{n}\sum_{i=1}^{n}\frac{{\left(p_{i}-\frac{1}{n}\right)}^{2}}{\frac{1}{n}} \end{array}$$
The authors use a divergence they call the $\chi^{2}$ divergence (we note that other authors use this term for a number of different functions), which they define as
(19)$$\begin{array}{c} \chi^{2}\left(p:q\right)=\sum_{i=1}^{n}\frac{{\left(p_{i}-q_{i}\right)}^{2}}{q_{i}} \end{array}$$
and given this form, it can be seem that the definition of the MR is application of the $\chi^{2}$ divergence between p and $U_{n}$.

In the context of softmax applied to a vector of logits, it is evident that a higher temperature leads to a smaller MR and furthermore that each element becomes closer to $U_{n}$. We quantify this relationship for our various experimental datasets in Section 10. Figure 6 shows the absolute values for our four datasets at different temperatures.

One of the results in this paper shows an approximate equivalence between the MR and the Kullback–Leibler divergence over the same operands, i.e.,
(20)$$\begin{array}{c} MR\left(p\right)=\chi^{2}\left(p:U_{n}\right)\approx KLD\left(p,U_{n}\right) \end{array}$$
with this approximation becoming ever closer as the value p becomes closer to $U_{n}$, that is, as the measure of roughness of p decreases, all three of these measures become more similar.

Finally, we note that $\chi^{2}$ is an asymmetric divergence measure, but can be used to define a symmetric divergence in a similar manner as the KLD is used to define the JSD:(21)$$\begin{array}{c} S\chi^{2}\left(p,q\right)=\chi^{2}\left(p:\frac{p+q}{2}\right)+\chi^{2}\left(q:\frac{p+q}{2}\right) \end{array}$$
and in fact this divergence is equal to triangular divergence (Equation (5)) after the summed terms are factored out. In the next section, we explain an almost perfect correlation between triangular divergence and Jensen–Shannon divergence.

This is not quite sufficient, as the proof shows the correspondence between these divergences when applied from the vectors p,q to $U_{n}$, rather than $\frac{p+q}{2}$. With high temperatures, however, these entities become very close. Given the experimental evidence of the very strong correlation, we believe this explanation sheds considerable light on the reason for the observation.

## 7. Correlation: Jensen–Shannon Divergence to Triangular Divergence

In [4], Topsøe shows a strong relationship between Jensen–Shannon divergence and triangular divergence in terms of an upper bound:(22)TRI(q,p)≤JSD(q,p)≤log2·TRI(q,p)

This is an encouraging result to start with but does not go far enough to explain the experimental correlations we measure, which show the two functions to be in almost perfect correlation over high-dimensional spaces.

First, we show a rewrite of the triangular divergence:(23)$$\begin{array}{cc} TRI(q,p)\; & =\;\frac{1}{2}\sum_{i=1}^{n}\frac{{\left(q_{i}-p_{i}\right)}^{2}}{q_{i}+p_{i}}\\ \; & =\;\frac{1}{2}\sum_{i=1}^{n}\frac{{\left(q_{i}+p_{i}\right)}^{2}-4p_{i}q_{i}}{q_{i}+p_{i}}\\ \; & =\;1-\sum_{i=1}^{n}\frac{2p_{i}q_{i}}{q_{i}+p_{i}}\;\mathrm{as}\;\sum_{i=1}^{n}q_{i},\sum_{i=1}^{n}p_{i}=1 \end{array}$$
We repeat our definitions of the CED and JSD from above:(24)$$\begin{array}{cc} JSD(q,p)\; & =\;1-\frac{1}{2}\sum_{i=1}^{n}h\left(q_{i}\right)+h\left(p_{i}\right)-h\left(q_{i}+p_{i}\right) \end{array}$$(25)$$\begin{array}{cc} CED(q:p)\; & =\;-\sum_{i=1}^{n}q_{i}\log p_{i} \end{array}$$
which allow us to note the following:There is now a strong apparent congruence between the TRI and JSD, based on the approximate equivalence of the component terms
(26)$$\begin{array}{c} h\left(q_{i}\right)+h\left(p_{i}\right)-h\left(q_{i}+p_{i}\right)\approx \frac{2p_{i}q_{i}}{q_{i}+p_{i}} \end{array}$$Note that these terms act as “similarity accumulators” in their respective contexts, and the approximate equivalence also implies that the respective divergence functions will yield similar values. This is not a strongly bounded equivalence but pragmatically holds when $q_{i}$ and $p_{i}$ are in the typical range of values we consider. If $q_{i}=p_{i}$, then both terms are equal to $2q_{i}$.Considering the evaluation time, the CED requires, for each vector dimension, a log calculation and a multiplication operation, whereas the TRI requires an addition, a multiplication, and a division. These operators are much cheaper for conventional hardware than the expensive log calculation. Over various data, we measured the relative cost as between 2 and 20 times different. Section 10 gives some actual times as measured over the different datasets used in this article.

### Mathematical Rationale of the Correlation

In [4], Topsøe introduces an ordered set of triangular divergence functions
(27)$$\begin{array}{c} TRI_{v}\left(q,p\right)=\frac{|q_{i}-p_{i}{|}^{2v}}{{\left(q_{i}+p_{1}\right)}^{2v-1}} \end{array}$$
where *v* is a natural number, and this is used to provide a perfect equality with the JSD:(28)$$\begin{array}{c} JSD\left(q,p\right)=\sum_{v=1}^{\infty }\frac{1}{2v(2v-1)}TRI_{v}\left(q,p\right) \end{array}$$
noting that the first term v=1 gives the TRI as in Equation ().

Clearly, the factor $\frac{1}{2v(2v-1)}$ decays quickly as *v* increases, leading to convergence as long as the $TRI_{v}$ function also decreases. What is also evident is that, as the measure of roughness of q and p decreases, then the numerator term $|q_{i}-p_{i}{|}^{2v}$ very rapidly diminishes to zero, while the denominator decreases much less quickly. The overall effect is that the first term, where v=1, becomes fully dominant in the summation, giving the required result of TRI(q,p)≈JSD(q,p).

Figure 7 shows this effect between two randomly selected GoogleNet vectors at temperatures of 1, 5, and 10. It can be seen that, as temperature increases, the $Tri_{1}$ term completely dominates the summation.

## 8. Recap: Correlation of Cross-Entropy and Triangular Divergence

The correlation between cross-entropy follows directly from the results above and is shown earlier in the paper in Figure 4. It can be observed that when *t* is increased to a value of 10 or more, the correlation between the two functions becomes essentially perfect. We have not included the corresponding diagrams for AlexNet, SqueezeNet, and Dinov2 since they essentially show the same effect.

## 9. Correlation of Cross-Entropy and Euclidean Divergence

Some modern networks, rather than classifying the input into a number of categories, instead aim to provide the post-trained logit space as an embedding which can either be used as the basis for further classification or else used as a similarity space in its own right. That is, for the universal set of possible output logits *U*, there should exist a dissimilarity space (U,d) with the property that, for any $u_{i},u_{j},u_{k}\in U$, $d\left(u_{i},u_{j}\right)<d\left(u_{i},u_{k}\right)$ implies that the input object resulting in $u_{i}$ should be more similar to that resulting in $u_{j}$ than the one resulting in $u_{k}$. Within our example data, DinoV2 is such a network.

The properties of the function *d* are required to be very different from cross-entropy: in any search domain it would be expected that *d* has at least semi-metric properties, including $d(u_{i},u_{j})\geq 0$, d(u,u)=0 and $d\left(u_{i},u_{j}\right)=d\left(u_{j},u_{i}\right)$. Depending on the search mechanism to be used, it may also be important that *d* is a proper metric, therefore also requiring the triangle inequality to be shown. As our discussion to this point has been entirely based on correlation, none of these properties has featured so far.

In the DinoV2 paper [12], the authors describe training the network using primarily cross-entropy and then test it using cosine distance over the logits. The reason for this is not stated but may perhaps be simply that cosine distance is a proper metric, is cheap to evaluate, and gives apparently good results. Other previous work has also suggested the use of Euclidean distance; and [19] develops a specialised metric based on cosine distance for a particular purpose. We have not, however, seen any principled argument for the metric of choice, one reason perhaps being that in high-dimensional spaces, most metrics are reasonably well correlated, and it is very challenging to tell which metric is semantically the best over a very large metric space for which no ground truth can feasibly be constructed.

We note first that triangular divergence as defined in Equation (5) is the square of a proper metric, which leads to the possibility of using the post-softmax space with this metric. As far as we know, this is a completely novel idea, and we are currently investigating it further.

We have observed one last very strong correlation, which is between Euclidean distance in the logit space and cross-entropy in the space to which softmax has been applied. The correlation is much tighter than that of cosine distance in the logit space, and leads to the suggestion that Euclidean distance may be the better metric to use in the case where logits are exported for use in the context of similarity search.

Figure 8 shows correlations, in the DinoV2 context, between Euclidean distance over the raw logit values and the CED over the softmax values for a range of temperatures. Figure 9 shows the same correlations for cosine distance, where $COS\left(u,v\right)\;=\;EUC\left(\frac{u}{{∥u∥}_{2}},\frac{v}{{∥vs.∥}_{2}}\right)$. It can be seen that these correlations are much weaker, leading to the suggestion that, for a such a network trained using cross-entropy, then Euclidean distance may be the better choice.

We have observed these correlations also in other spaces, but we note that they do not hold for all Euclidean and cosine spaces. We do not yet fully understand the properties necessary to achieve these strong correlations, nor a full mathematical basis for them, and for the moment we leave this as an item of further work.

## 10. Discussion

### 10.1. Cost of CED and TRI Application

Table 3 shows simple measurements of the CED and TRI functions, showing that the TRI is cheaper to evaluate at least in this context of measurement. It is of course impossible to provide objective measurement as the cost will depend on many features of the hardware and software context.

In this case, we used MATLAB 2024a (which is optimised for the M1 chipset) running on a MacBook M1 Pro with 32G of main memory. The MATLAB functions measured are as follows:
    CED = @(X,Y) - sum(X .* log(Y),2);    TRI = @(X,Y) - sum((X .* Y) ./ (X + Y),2);
thus using the optimisation of the TRI shown in Section 7. Note that these forms take arrays of data, rather than a single datum, as input. Timing was performed using the MATLAB timeit call over a lambda form which applies each function to a single datum against 10k others. All tests were repeated until the standard error of the mean was less than 1% of the mean values reported.

As can be seen, and also as expected, the TRI function is always significantly less costly than the CED. We are aware that in the context of machine learning, this cost may not be significant to the overall training time, but in cases where the correlation is almost perfect, we see no good reason to use extra compute cycles.

### 10.2. Temperature and Measure of Roughness

Guided by figures we derived from [1], we started on our experiments, applying temperature values in the range 0.1 to 100, and observed the very tight correlations with values of around 10 or greater over the different datasets used.

Having subsequently discovered the underlying mathematical relations based on the measure of roughness, we believe that is the more principled concept from which the correlations derive. Figure 6 shows how this varies with the application of temperature, and a correlation can be seen between the individual graph for each dataset and the properties of the logit range and magnitude shown in Table 2.

In [1], the suggestion is for researchers to experiment across the range of temperatures; we suggest experimenting across temperatures which achieve the MR down to a value of, for example, $10^{-6}$, which may give a more useful range of temperatures with which to experiment. Notice that even with our small number of datasets, this implies very different temperatures.

## 11. Conclusions and Further Work

In this article, we have shown a number of very strong correlations between the cross-entropy divergence function and other information distances. Cross-entropy is almost ubiquitously used in the training of neural networks. These correlations are interesting in their own right and have one potential practical application in the correlation between cross-entropy and triangular divergence, as the latter is substantially cheaper to evaluate and should perhaps be preferred in cases where the correlation is almost perfect.

We further show a more surprising, and as yet not fully explained, correlation between Euclidean distance in the logit space and cross-entropy in the post-softmax space. We suggest this may imply that, where network embeddings are exported for use in more general similarity spaces, Euclidean distance may be the metric of choice, as opposed to cosine distance which seems to be more commonly used.

Three items of further work are compelling as a result of this work. First, we observe that, rather than using either Euclidean or cosine distance in the logit space for the purpose of general similarity, it is equally possible to use a proper metric form of triangular distance in the post-softmax space. We are currently investigating this and have observed some possible advantages with respect to technical properties of the resulting space.

Second, the very strong correlation between Euclidean distance in the logit space and the CED in the post-softmax space does not apply to all Euclidean spaces, and we do not as yet have a full understanding of the properties required in the Euclidean space or consequently a mathematical explanation of the correlation.

Finally, we would like to test whether cosine or Euclidean distance over the logits does indeed give a better semantic test over the input space. Such testing is very challenging over very large input spaces, as it is impossible to construct a meaningful ground truth due to the quadratic number of assessments required; we are working on an approximate measure of quality with a view to achieving such comparisons. 

## Figures and Tables

**Figure 1 entropy-26-00491-f001:**
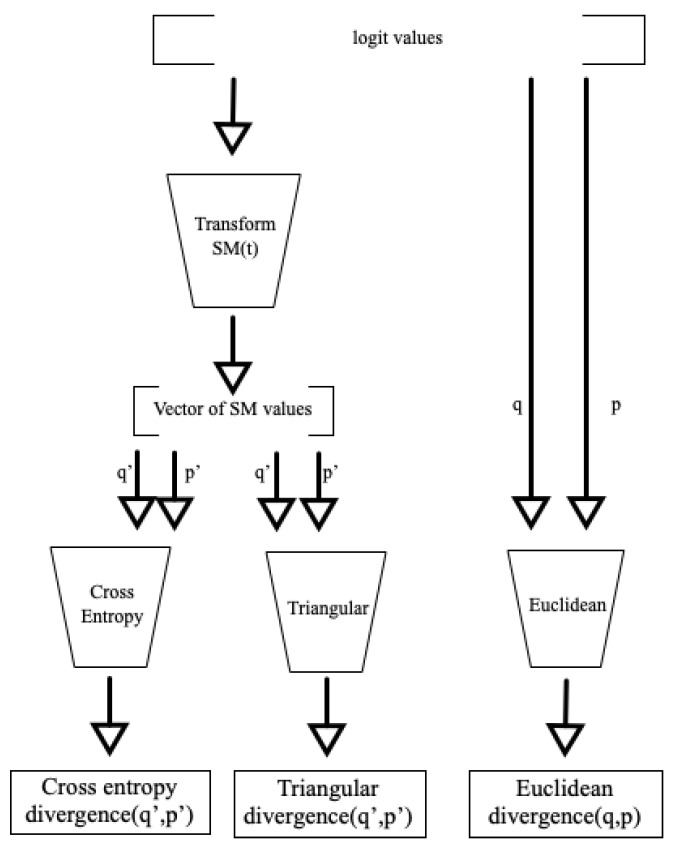
The three main correlations shown in this article. For higher temperature values, there is typically an almost perfect correlation between cross-entropy divergence and triangular divergence. In some spaces, these also correlate very strongly with Euclidean divergence in the logit space.

**Figure 2 entropy-26-00491-f002:**
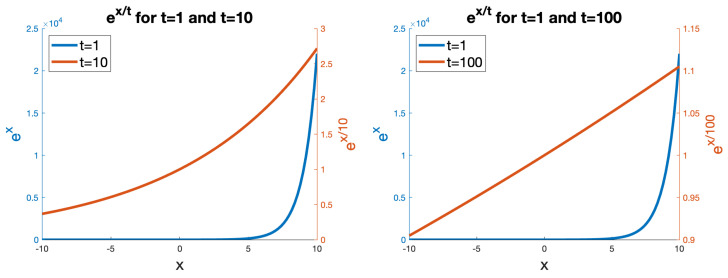
The effect of different temperature values *t* (here t∈{1,10,100}) on the function $e^{x/t}$. Note that the absolute values are not significant, as $L_{1}$ normalisation occurs over the whole vector after this application.

**Figure 3 entropy-26-00491-f003:**
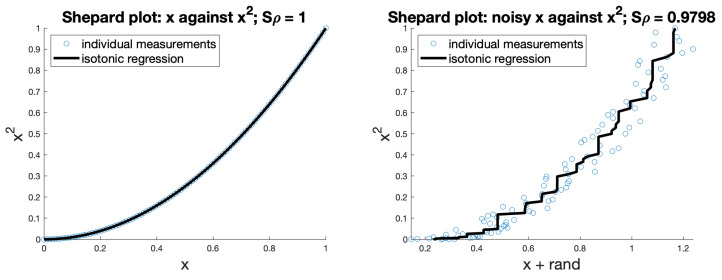
Demonstration of Shepard plots; the first shows that *x* and $x^{2}$ are perfectly correlated, the second adds some random noise to *x* to show an imperfect correlation.

**Figure 4 entropy-26-00491-f004:**
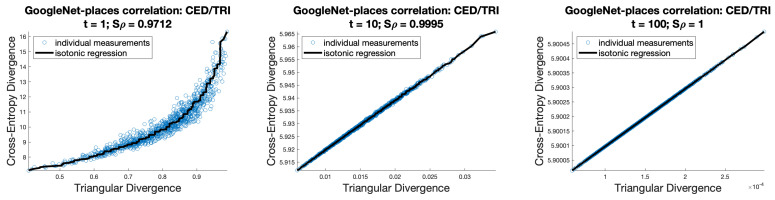
Correlations for GoogleNet-Places logits. For three different values of softmax temperatures—1, 10, and 100—we see how the correlation between cross-entropy and triangular divergence becomes essentially perfect as temperature increases.

**Figure 5 entropy-26-00491-f005:**
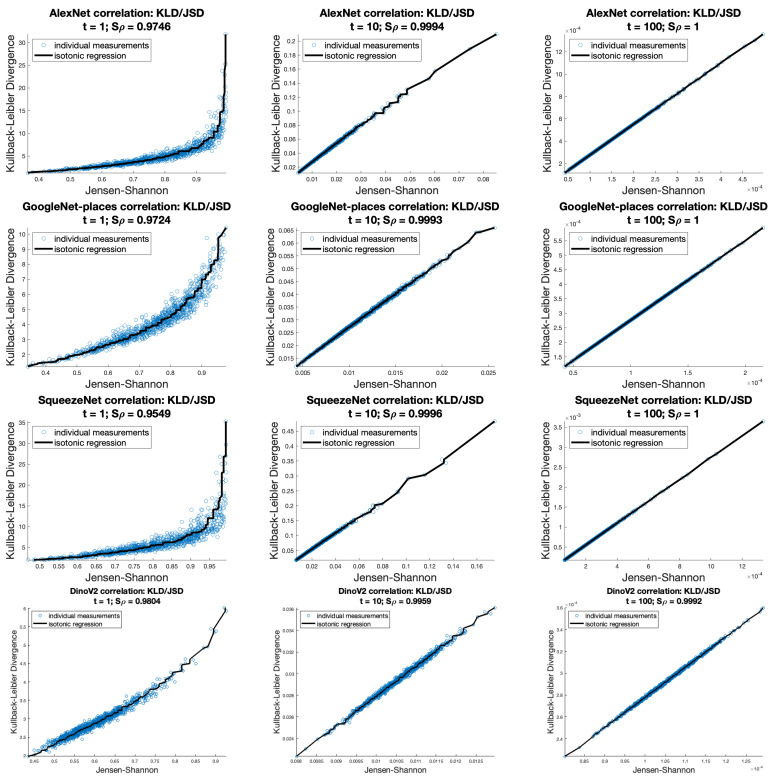
Shepard plots and Spearman’s rho correlation between the KLD and JSD across AlexNet, GoogleNet, SqueezeNet, and DinoV2 datasets, with variations in softmax temperature t∈{1,10,100}.

**Figure 6 entropy-26-00491-f006:**
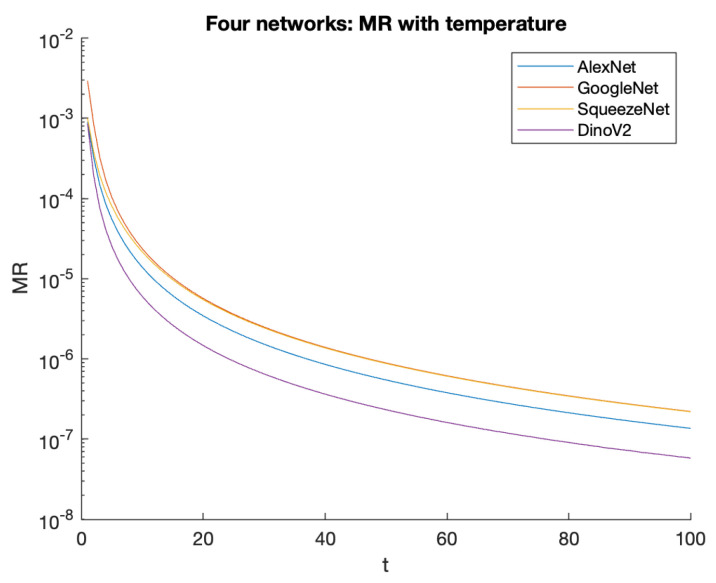
The relation between temperature and the measure of roughness for the four datasets considered. Note the logarithmic scale of the Y axis. Although all graphs show a similar shape, note that very different temperatures may be required to achieve the same MR.

**Figure 7 entropy-26-00491-f007:**
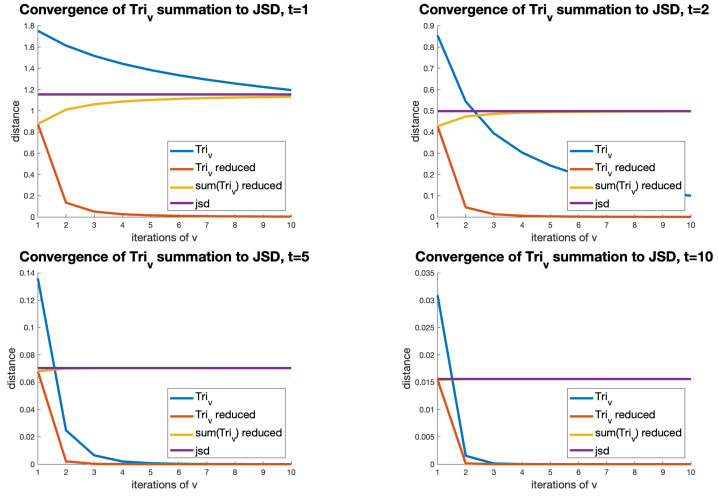
The summation of $Tri_{v}$ terms for v=1,…,10 between two GoogleNet vectors, with temperatures of 1, 2, 5, and 10. The four lines in the plots represent the following: **${\mathrm{Tri}}_{v}$**: the $TRI_{v}$ formula at different values of *v*; **${\mathrm{Tri}}_{v}$ reduced**: the $Tri_{v}$ value adjusted by the factor $\frac{1}{2v(2v-1)}$; **Sum (${\mathrm{Tri}}_{v}$) reduced**: the sum of these terms up to this value of *v*; and **jsd**: the (constant) outcome of the JSD function. As temperature increases, it can be seen how the adjusted $Tri_{1}$ term increasingly dominates the summation, becoming indistinguishable from the JSD at v=1 and t=10.

**Figure 8 entropy-26-00491-f008:**
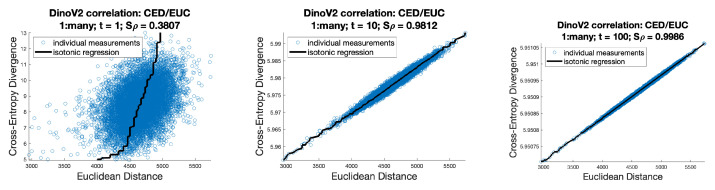
Shepard plots and Spearman’s rho correlation between the CED (over the softmax’d value) and EUC (over the raw logit values) for DinoV2 dataset, with variations in softmax temperature t∈{1,10,100}.

**Figure 9 entropy-26-00491-f009:**
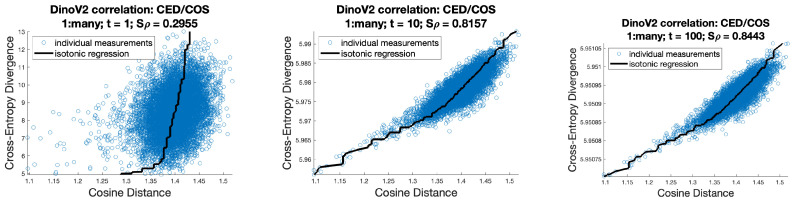
Shepard plots and Spearman’s rho correlation between the CED (over the softmax’d value) and COS (over the raw logit values) for DinoV2 dataset, with variations in softmax temperature t∈{1,10,100}.

**Table 1 entropy-26-00491-t001:** Outline description of the functions of interest. As we are only interested in correlations, none of these are proper (metric) distances; the JSD, TRI, and EUC are the squares of proper distances. The CED and KLD are asymmetric in their arguments, and the others are symmetric.

CED	Cross-entropy	Our main topic of interest, as applied to the output layer of networks after softmax
KLD	Kullback–Leibler divergence	A principled information loss function
JSD	Jensen–Shannon divergence	A “smoothed, symmetrised” version of the KLD
TRI	Triangular divergence	A little-known divergence with tight bounds over the JSD. The square root of this form is also sometimes referred to as chi-square distance, although that term is also used for other functions
EUC	Euclidean divergence	Euclidean divergence, the square of the classic $L_{2}$ distance

**Table 2 entropy-26-00491-t002:** Data used for experiments. Magnitudes given are mean, measured from the data centroid.

Network Name	Training Dataset	Logit Name	Dimensions	Range	Magnitude
GoogleNet	Places365	loss3-classifier	365	[−8.9,20.9]	41.7
SqueezeNet	ImageNet	pool10	1000	[0,58.5]	105.8
AlexNet	ImageNet	fc8	1000	[−13.6,43.4]	77.4
DinoV2	n/a	n/a	384	[−14.0,14.2]	46.4

**Table 3 entropy-26-00491-t003:** Evaluation cost of the CED and TRI for different networks. The values reported are seconds per divergence calculation.

Network Name	CED Cost	TRI Cost
GoogleNet	$2.4\times 10^{-6}$	$2.0\times 10^{-7}$
SqueezeNet	$1.4\times 10^{-6}$	$5.2\times 10^{-7}$
AlexNet	$4.1\times 10^{-6}$	$5.3\times 10^{-7}$
DinoV2	$2.2\times 10^{-6}$	$1.3\times 10^{-7}$

## Data Availability

We have made available all data and code used to perform the experiments described here at https://github.com/MetricSearch/2024_entropy_paper (accessed on 29 May 2024).

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
