# Peer review of "Correlations of Cross-Entropy Loss in Machine Learning"

_entropy, 2024, doi:10.3390/e26060491_

Round 1
Reviewer 1 Report
Comments and Suggestions for Authors
This work studies the correlations between cross-entropy and other popular losses in training deep neural networks (DNNs). The correlations are mostly empirical observations, especially under a large value of the temperature hyperparameter of cross-entropy. These observations may lead to better training loss for DNNs.
Weaknesses of this work:
1. The main claims are empirical. It is not new that, by varying the temperature hyperparameter, the cross-entropy loss can become sharp or flat to mimic linear or quadratic functions.
2. The correlations revealed by this work can be mathematically formulated based on a generalized version of softmax like in [1], rather than purely empirical. I just randomly searched the literature, the authors may find more inspirations from other forms of generalization.
3. It seems this paper is closely related to [2], where they show all these divergence measures can be uniformly represented by a geometric measure (local intrinsic dimensionality) for tail distributions.
4. The use of cosine similarity is because it is magnitude-free and is more robust to outliers than square loss. It is also because the deep representations are more geometrically meaningful in the latent space than the Euclidean distance. The other factor that needs to be considered is the stability of the gradient when a loss function is adopted. I hope the authors can discuss these practical considerations in the revision.
[1] Gao, Yingbo, et al. "Exploring kernel functions in the softmax layer for contextual word classification." arXiv preprint arXiv:1910.12554 (2019).
[2] Bailey, James, Michael E. Houle, and Xingjun Ma. "Local intrinsic dimensionality, entropy and statistical divergences." Entropy 24.9 (2022): 1220.
Comments on the Quality of English Language
There are places hard to follow without introducing the necessary background.
Author Response
Entropy, submission: entropy-3017950
Report on changes made after first review
We thank the reviewers kindly for their suggestions for improvement of the paper. It was clear we had not quite identified very clearly the main subject, which is the main changes we have made. Also we have made it clearer what part of the work was included as background and related, and what part is the novel contribution, which we hope can help.
Enclosed are more detailed responses:
Review 1:
This work studies the correlations between cross-entropy and other popular losses in training deep neural networks (DNNs). The correlations are mostly empirical observations, especially under a large value of the temperature hyperparameter of cross-entropy. These observations may lead to better training loss for DNNs.
Weaknesses of this work:
- The main claims are empirical. It is not new that, by varying the temperature hyperparameter, the cross-entropy loss can become sharp or flat to mimic linear or quadratic functions.
We have added a sentence in the Introduction to address the empirical concern, and made it clearer that we report the linearity of high-temperature softmax function as background work rather than a contribution.
- The correlations revealed by this work can be mathematically formulated based on a generalized version of softmax like in [1], rather than purely empirical. I just randomly searched the literature, the authors may find more inspirations from other forms of generalization.
We have made it clearer that we provide explanations based on mathematical literature which we cite; we cannot see the relevance of citation [1] which uses different kernel function for logit calculation and the inner product as a loss function; our citation Agarwala et al. (ref [1] in the manuscript) is to more recent Google Brain research on the variance of temperature.
- It seems this paper is closely related to [2], where they show all these divergence measures can be uniformly represented by a geometric measure (local intrinsic dimensionality) for tail distributions.
We have put in a reference to [2]in the Background and Related Work section. We were aware of this paper but did not see a close relation to our work, as that is about distributions of distances within infinitesimal regions of the space, rather than correlations over the wide space. - The use of cosine similarity is because it is magnitude-free and is more robust to outliers than square loss. It is also because the deep representations are more geometrically meaningful in the latent space than the Euclidean distance. The other factor that needs to be considered is the stability of the gradient when a loss function is adopted. I hope the authors can discuss these practical considerations in the revision.
Thank you for this explanation. As the topic is rather beyond the purpose of our contribution, and we do not wish to confuse the readers, we have consequently removed the reference to any possible reason for the use of cosine distance. We believe the observation that the strong correlation between Euclidean and cross-entropy stands without further exposition.
[1] Gao, Yingbo, et al. "Exploring kernel functions in the softmax layer for contextual word classification." arXiv preprint arXiv:1910.12554 (2019).
[2] Bailey, James, Michael E. Houle, and Xingjun Ma. "Local intrinsic dimensionality, entropy and statistical divergences." Entropy 24.9 (2022): 1220.
Comments on the Quality of English Language
There are places hard to follow without introducing the necessary background.
We have thoroughly reviewed the text and made a number of improvements in this respect.
Review 2:
The manuscript studies different divergencies in the context of machine learning. It is difficult to assess the contribution of this study in the field since the research problem is not formulated and the paper does not have any substantial literature review. It appears that the main contribution of the manuscript is based on an obvious observation that the information divergence functions being considered (which include Cross-entropy, Triangular and Euclidean) are highly correlated. This is a foregone conclusion, particularly, because the Spearman ranked correlation is used in the paper. It is well-known that a Spearman correlation equals 1 when the two compared variables are monotonically related (even if their relationship may not be linear). In the other words, the paper simply illustrates the following fact: the further apart (in some sense) the two compared variables (or points in a space) are from each other, the greater value of an information divergence function.
We have added some introductory sentences and changed the abstract to make the contribution clearer. We do not think that these correlations are obvious, and they have not been previously noted. We have been working with f-divergences for some 20 years and were amazed at the closeness of the correlations we observed. Some *bounds* have been previously noted, which we cite. We understand the Spearman rho measure to be the gold standard measurement of correlation and do not understand this criticism. The main point of the paper is to quantify the “in some sense” referred to by the reviewer; this “sense” is captured by the various information divergence functions which are of course all different and generally are not highly correlated. We hope our new words in the abstract and introduction make this point clearer.

Reviewer 2 Report
Comments and Suggestions for Authors
The manuscript studies different divergencies in the context of machine learning. It is difficult to assess the contribution of this study in the field since the research problem is not formulated and the paper does not have any substantial literature review. It appears that the main contribution of the manuscript is based on an obvious observation that the information divergence functions being considered (which include Cross-entropy, Triangular and Euclidean) are highly correlated. This is a foregone conclusion, particularly, because the Spearman ranked correlation is used in the paper. It is well-known that a Spearman correlation equals 1 when the two compared variables are monotonically related (even if their relationship may not be linear). In the other words, the paper simply illustrates the following fact: the further apart (in some sense) the two compared variables (or points in a space) are from each other, the greater value of an information divergence function.
Author Response

(The authors gave the same response as above.)

Round 2
Reviewer 2 Report
Comments and Suggestions for Authors
N/A.